# All-printed large-scale integrated circuits based on organic electrochemical transistors

Peter Andersson Ersman[1]*, Roman Lassnig[1], Jan Strandberg[1], Deyu Tu [2], Vahid Keshmiri[3],
Robert Forchheimer[3], Simone Fabiano [2,4]*, Göran Gustafsson[1] & Magnus Berggren[2,4]*

The communication outposts of the emerging Internet of Things are embodied by ordinary items, which desirably include all-printed flexible sensors, actuators, displays and akin organic electronic interface devices in combination with silicon-based digital signal processing and communication technologies. However, hybrid integration of smart electronic labels is partly hampered due to a lack of technology that (de)multiplex signals between silicon chips and printed electronic devices. Here, we report all-printed 4-to-7 decoders and seven-bit shift registers, including over 100 organic electrochemical transistors each, thus minimizing the number of terminals required to drive monolithically integrated all-printed electrochromic displays. These relatively advanced circuits are enabled by a reduction of the transistor footprint, an effort which includes several further developments of materials and screen printing processes. Our findings demonstrate that digital circuits based on organic electrochemical transistors (OECTs) provide a unique bridge between all-printed organic electronics (OEs) and low-cost silicon chip technology for Internet of Things applications.

[1] RISE Acreo, Department of printed electronics, Bredgatan 33, SE-602 21 Norrköping, Sweden. [2] Laboratory of organic electronics, Department of science and technology, Linköping University, SE-601 74 Norrköping, Sweden. [3] Information Coding Group, Department of electrical engineering, Linköping University, SE-581 83 Linköping, Sweden. [4] Wallenberg Wood Science Center, Linköping University, SE-601 74 Norrköping, Sweden. *email: peter.andersson.ersman@ri.se; simone.fabiano@liu.se; magnus.berggren@liu.se

Printed organic electronics (OE), in the form of sensors[1–3], actuators[4–6], and displays[7–9], have been manufactured using printing techniques on flexible or stretchable carriers to enable distributed monitoring and safety functions on packages, in smart labels, in artificial skin applications, and more[10–12]. Organic solid-state semiconducting devices, relying on charge injection or field-effect charge modulation, represent one important class of relevant devices for internet-of-things (IoT) applications[13]. These are typically built up from configurations including (ultra-)thin films and/or very narrow distances between electrodes; especially if low-voltage and fast operation is targeted[14–17]. Organic field-effect transistors (OFETs) have been explored as sensors and also in various large-scale integrated circuits and they typically operate at voltages beyond 10 V[18–20] and at frequencies ranging from $10^3$ to $10^6$ Hz[21–23], thus making relatively swift sensing and signal processing possible. One major drawback with all-printed OFETs, besides high-voltage operation and challenging device structure, is instability or drift of the fundamental threshold voltage[24–26].

For IoT labels that require massive and reliable signal processing, one or several silicon (Si) chips, typically operating in the range 1.2–3.3 V, are utilized in combination with peripheral, sometimes printed, devices for sensing, actuation, and interfacing[27]. These chips are archetypally assembled and connected onto the label or item using flip-chip mounting or wiring techniques. The cost of Si-chips is to a great part dominated by the total area of, and the number of process steps to manufacture, the actual Si-dice. For multifunctional applications, several dedicated devices encircle the Si-chip and each requires a unique set of contact pads along the Si-chip (Fig. 1a), which will increase chip area and thus drive costs. In fact, for application-specific integrated circuits (ASIC), such as IoT chips, already beyond just a few contact pads, the chip area is primarily dominated by the number of contact pads, i.e., the design is pad limited rather than core limited. This problem becomes even worse considering that improved silicon manufacturing processes mainly enable core miniaturization, rather than pad miniaturization[28]. Hence, minimizing the chip area required by the contact pads is of great importance, as this will lower the manufacturing costs (Fig. 1b). Pad limited chip designs typically lead to expensive multifunctional IoT labels, which hampers the implementation of smart labels for desired multipurpose and end-user specific applications in medicine, safety, and security. With printed digital circuits, such as shift registers and binary-coded decimal (BCD) decoders that transfer serial or binary-coded signals into exclusive output terminals, low-cost multifunctional IoT labels based on Si-chips that only need few contact pads would then be feasible.

Organic electrochemical transistors (OECTs) represent another class of printable devices that typically operate at ~1 V, with exceptional threshold voltage stability and high transconductance characteristics[29–31]. These features, along with very robust device architecture and biocompatibility, have made the OECTs suitable for sensing and various bioelectronics applications; as the translator of chemical, physical, and biochemical signals into electrical ones, across the biology-technology gap[32–35]. The prime drawbacks of these devices are the relatively slow switching characteristics, in part owing to diffusion of ions, and the need for an electrolyte in the gate configuration; the latter makes miniaturization difficult owing to a relatively large footprint of printed electrolyte features. Few attempts have been reported to realize OECT-based digital circuits[36–39], and the efforts have typically been limited to individual digital gates and active addressing in display matrices[40–42]. Slow switching and a large device footprint, i.e., on the scale of 10–100 ms and 0.1–1 mm$^2$, respectively, which is archetypical for printed OECTs, is not suitable for signal processing but certainly tolerable in decoders, shift registers, and multiplexers. Particularly, this is valid when driving and recording already slow operating and areal devices, such as electrochromic displays, sensors, and actuators, which typically switch and operate at 0.1–10 Hz and are of a size ranging from 0.1 mm$^2$ to 1 cm$^2$.

Here, we report all-printed coplanar OECTs included in a BCD 4-to-7 decoder and a seven-bit shift register, connected to Si-circuitry, to drive a monolithically integrated all-printed electrochromic seven-segment display, based on poly(3,4-ethylenedioxythiophene) doped with poly(styrenesulfonic acid) (PEDOT:PSS). The BCD 4-to-7 decoder, consisting of 87 OECTs, operates as a four-stage combination logic function and reduces the number of pins required on the Si-chip, to drive the display, from 7 to 4. To even further reduce the number of contact pins on the Si-chip, a seven-bit serial-in/parallel-out shift register, which requires only one data input and one clock signal, was designed and printed. The seven-bit shift register consists of seven cascaded Master-Slave D flip-flops based on 114 OECTs in total, thus requiring seven clock cycles to display one digit. Both the shift registers and the BCD decoders are implemented with NAND/NOT logic circuits and are based on the all-printed coplanar OECTs. The NAND/NOT logic implementation relies on a resistor ladder and suffers from logic level decay, owing to the particular characteristics of the PEDOT:PSS-based OECT that operate in depletion

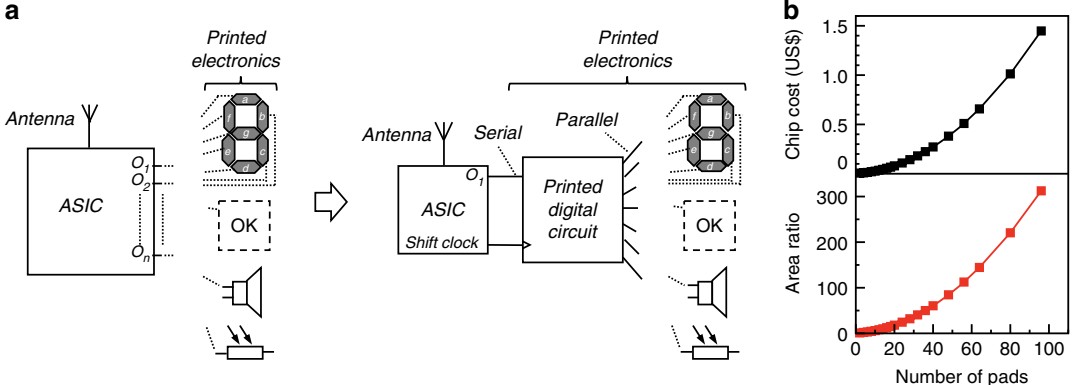

**Fig. 1** Concept of hybrid electronics systems combining printed electronics and silicon chips. **a** Application-specific integrated circuits (ASIC) require a large number of pads to implement multiple functionalities. Printed digital circuits can significantly reduce the number of pads, hence, the twinning of printed electronics and Si-based electronics into hybrid electronics systems enable ultra-low-cost applications. **b** The non-linear growths of ASIC chip processing cost and required chip area as the number of pads increases; the cost and area estimations are based on an 8-inch silicon wafer at a processing cost of 1500 US$, a pad area of 0.01 mm$^2$ and a pad pitch of 0.15 mm

mode. However, here we successfully demonstrate correct logic propagation through 29 stages of OECT-based gates in the seven-bit shift registers without significant voltage level decay. This was obtained by increasing the margins of the voltage output levels, enabling cascade-coupling of the logic gates. An increase in voltage margins also implies an elevated risk for device degradation, owing to increased voltage strain between the gate and drain. To lower the voltage strain, the circuit supply voltage was ± 5 V, with a corresponding matching of the resistor values of the ladder. Moreover, a low-voltage strain implies better OECT performance, i.e., lower OFF-current levels, which in turn results in better defined output levels when constructing logic circuits. We further demonstrate the robustness of this technology by monolithically integrating the BCD 4-to-7 decoder with a seven-segment electrochromic display, where screen printing is the sole deposition technique used to manufacture the electronic system. The possibility to develop all-printed large-scale ICs on flexible substrates, operated at low voltages, offers an unreached solution when targeting flexibility in multipurpose applications.

## Results

**Printed organic electrochemical transistors.** The PEDOT:PSS-based OECTs are printed to form a coplanar top-gate configuration (Fig. 2a, b), where a printed solid electrolyte is sandwiched between a bottom channel and a top-gate electrode, allowing for shorter switching times and smaller footprints, as compared with the lateral configuration[43]. Previously, we have presented two designs of coplanar OECTs, one comprising carbon source/drain electrodes in contact with the electrolyte (denoted as C-C OECT), and one where the carbon source/drain electrodes are isolated from the electrolyte (denoted as P-P OECT)[43]. The C-C OECTs offer larger ON-current and better switching cyclability, owing to the source and drain carbon contacts suppressing reduction front propagation in the PEDOT:PSS outside the active channel area[43]. Here, all the logic gates and digital circuits are based on the C-C OECT configuration. The characteristics of a typical printed C-C OECTs are presented in

Fig. 1c, d. The ON-current is ~0.5 mA and the ON/OFF ratio is ~$10^5$, whereas the switching time is ~20 ms (ON-to-OFF) and ~30 ms (OFF-to-ON), respectively.

**Logic gates based on printed electrochemical transistors.** Inverters and NAND gates serve as the building blocks of various combinational and sequential OECT-based logic circuits. We have manufactured and tested inverters, two-, three-, and four-input NAND gates, which are the building blocks of the BCD 4-to-7 decoder and the seven-bit shift register, and the results are presented in the following sections. The logic gates are based on OECTs operated in depletion mode and their output voltage levels, upon switching, therefore rely on voltage division of a resistor ladder ($R_1$, $R_2$, and $R_3$). The OECT-based inverters and two-input NAND gates are presented in Fig. 3a, b, whereas three- and four-input NAND gates are described in Supplementary Figs. 1–4. The selection of resistor values, and their mutual relationship, is dependent on the applied supply voltage and the targeted LOW and HIGH output voltage levels along with the resistance of the ON/OFF states of the OECT. The resistance of the OFF-state (ON-state) should preferably be at least ~20 times higher (lower) than $R_3$ to ensure reliable switching (Supplementary Fig. 5). Here, the target values of the resistor ladder are $R_1 =$ 40.5 k$\Omega$, $R_2 = 14$ k$\Omega$, and $R_3 = 30.5$ k$\Omega$. These values were obtained through simulations (Supplementary Fig. 6), and offer relatively fast switching with a small RC delay and a sufficient balance between $R_3$ and the resistance value of the corresponding OECT ON-state. Additionally, these values are chosen such that a resistance variation of ~10% would result in an output voltage variation of ~1% (Supplementary Fig. 6a). As shown in Fig. 3c–e, the HIGH and LOW output voltages of such logic gates are typically higher than 1.25 V and lower than 0.15 V, respectively, with LOW and HIGH input voltages of 0.1 V and 1.3 V. Such a small voltage difference between the input and output logic voltage levels ensures correct logic propagation in the cascade-coupled decoders and shift registers.

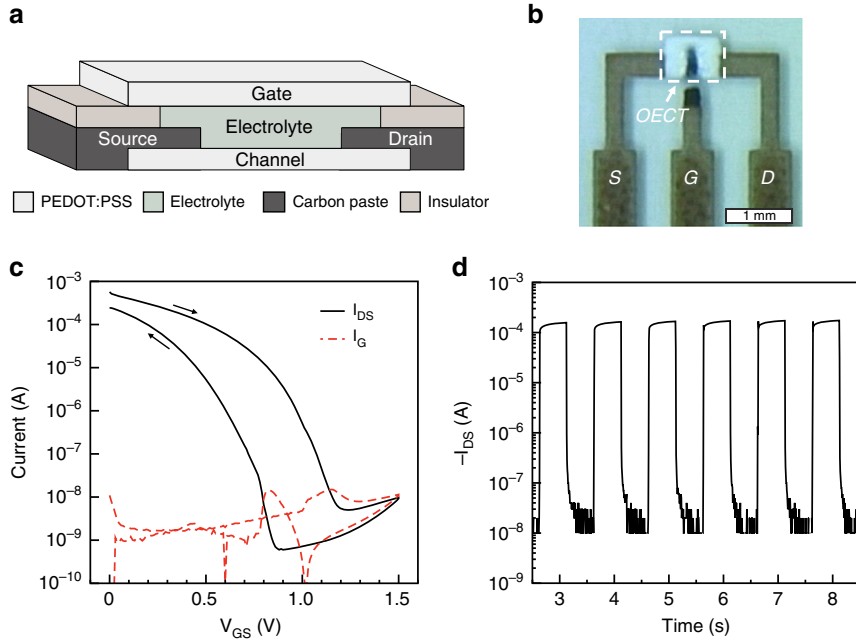

**Fig. 2** Printed organic electrochemical transistors. **a** The illustrated structure of an all-printed coplanar OECT-based on PEDOT:PSS. **b** Micrograph of an all-printed PEDOT:PSS-based OECT (scale bar: 1 mm). **c** Typical transfer characteristics of the printed PEDOT:PSS-based OECTs. **d** Typical switching characteristics of the PEDOT:PSS-based OECTs. A drain-source voltage ($V_{DS}$) of −1 V was used in **c** and **d**, and square wave pulses varying between −0.1 and 1.3 V at 1 Hz were used as gate voltages ($V_G$) in **d**

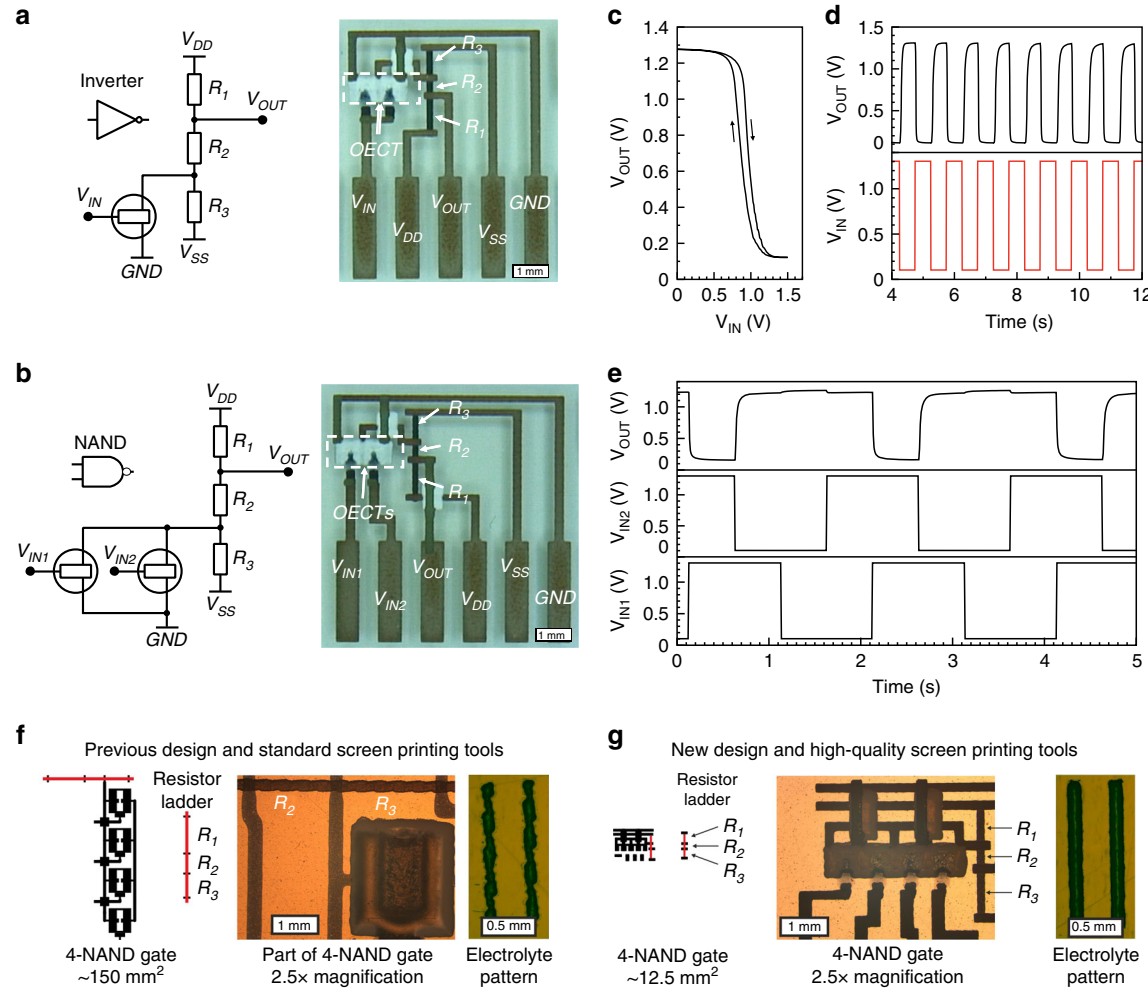

**Fig. 3** Logic gates based on printed electrochemical transistors. Schematics of an inverter (**a**) and a two-input NAND gate (**b**) based on OECTs, together with the corresponding optical images (scale bar: 1mm). The supply voltages $V_{DD}$ and $V_{SS}$ were +5V and −5V, respectively. **c** Voltage transfer characteristics and **d** switching characteristics of the inverter, where the propagation delay $t_p = 54$ ms. **e** Characterization of the two-input NAND gate, where the propagation delay $t_p = 20$ ms. **f** A four-input NAND gate according to the previous design (ref. [39]) requires an area of ~150 mm², partially owing to the relatively large area required for the resistor ladder. Microscope images of a portion of the four-input NAND gate (scale bar: 1mm) and an electrolyte pattern printed with a low-resolution screen mesh (scale bar: 0.5mm) are also shown, for comparison with the new design. **g** The four-input NAND gate according to the new design only requires an area of ~12.5 mm², which corresponds to a footprint reduction by a factor of ~12. This footprint reduction is explained by shrinkage of the resistor ladder area and utilization of high-resolution screen meshes. The four-input NAND gate manufactured according to the new design fits within the area of the microscope image (×2.5 magnification), whereas only one OECT, $R_2$, and $R_3$ from the previous design fit within the same microscope image area (scale bar: 1mm). In addition, the resistor ladder printed according to the new design is much more well-defined, which results in reproducible voltage output levels owing to higher matching accuracy between $R_1$, $R_2$, and $R_3$. The rightmost microscope image shows an electrolyte pattern printed with a high-resolution screen mesh, which results in well-defined printed features allowing for narrower design rules, further footprint reduction and improved manufacturing yield (scale bar: 0.5mm)

The operational and layout design requirements for the OECT technology, to serve as the signal switches in the targeted digital circuits, are ON-to-OFF and OFF-to-ON switching times shorter than 30 ms, an ON/OFF-current ratio of at least 400, and miniaturization of the OECT footprint down to ~1 mm². Together, this should allow for a digital circuit, comprised of ~100 OECTs configured in <30 logic stages, that fits within an area of ~10 cm² and that performs a complete signal processing task within a (few) second(s). To reach this desired combination of parameters, we redesigned the layout, materials, printing processes, and driving protocol of OECTs circuits, in several aspects, as compared with previous demonstrations[39,44]. Our earlier position with respect to line widths and layout is shown in Fig. 3f. In the current work, we have increased the sheet resistance (from ~5 kΩ sq⁻¹ to ~25 kΩ sq⁻¹) of the printed resistor material

by diluting the resistor ink with ~20% of dielectric ink (Supplementary Fig. 7), and also chosen to utilize a polyarylate screen mesh (offering better resolution) instead of the nylon meshes previously used. The high-resolution screen printing approach enables printing of better resolved features that gives a reduced footprint, which here includes a size reduction of the PEDOT:PSS-based channel, the electrolyte (Supplementary Fig. 8), the dielectric layer defining the electrolyte area and also the via interconnects. Thanks to the fivefold increase in sheet resistance of the resistor ink, the area of each resistor (200 µm linewidth) ladder is reduced by a factor of 5. A detailed description of all materials and printing processes is provided below in the Methods section. In addition, a supply voltage of ± 5 V was applied, which lowers the voltage strain between gate and drain. It is of critical importance to maintain the voltage

strain at a low level to obtain fully operational logic gates (Supplementary Fig. 9). The combined improvements, listed above, allows us then to reduce the total footprint of the OECT-based circuits by a factor 12–40 while comparing with prior reports[39,44] (Fig. 3g). Footprint reduction not only enables monolithic integration of printed logic circuits and electrochromic displays, at an adequate area ratio, it also implies an improved manufacturing yield, thanks to better deposition accuracy as the circuits are manufactured on a smaller area.

**BCD 4-to-7 decoder.** A BCD 4-to-7 decoder enables an update of a seven-segment display, e.g., a counting sequence from '0' to '9', by only using four input signals, instead of seven input signals when addressed directly. Here, the decoder, purely designed and based on OECT NAND gates and inverters, is presented in Fig. 4a, b. The decoder, which is a four-stage logic circuit, consists of 87 OECTs, 108 resistors, and >200 via interconnects. Figure 4c shows the logic voltage output sequence (red solid line, $V_{PE}$) that controls the update sequence of the pixel electrode of segment a (see inset to Fig. 4c for a definition of segments a–g in a seven-segment display) when the input signals $Data_{IN1}$ to $Data_{IN4}$ correspond to the digits [0–9]. The data shown by the black solid line is the logic voltage output sequence that controls the counter electrode of segment a ($V_{CE}$). For each digit, segment a shows the correct logic voltage level. All seven nodes a–g are tested to show correct functionality, more details can be found in the Supplementary Figs 10–15. The largest benefit of using the decoder is the small propagation delay, which is in total only 150 ms, whereas the seven-bit shift register presented below needs to process seven clock cycles, thus making it slower. The propagation delay was further investigated by cascade-coupling five printed OECT-based inverters into a ring oscillator circuit. After 60 s of continuous self-oscillations, the voltage output was varying between 0.55 V and 1.35 V at a frequency of ~1 Hz, which corresponds to ~100 ms propagation delay per stage of the ring oscillator circuit.

**Seven-bit shift register.** The attractive functional feature of a shift register is that only one input signal terminal, together with a clock signal, is required to produce a signal pattern on an arbitrary number of parallel output terminals. The key building blocks of shift registers are the D flip-flops (Fig. 5a), which transfer the input data at the D-terminal to the outputs Q and $Q_{INVERSE}$ for every clock cycle. A Master-Slave D flip-flop (Fig. 5a, b), based on OECTs, is built with eight two-input NAND gates. Cascading seven D flip-flop circuits, and adding two inverters to the first flip-flop circuit, results in a seven-bit shift register configuration (Fig. 5a–c), consisting of 114 OECTs, 174 carbon resistors, and over 400 via interconnects, which are all screen printed on a flexible substrate. Figure 5d presents the voltage outputs at nodes g and a, which are the first (Q1) and the seventh (Q7) parallel outputs in the shift register, respectively. In this test, the input signal Q is alternating between '0' and '1', and shifting of bits is triggered by a clock pulse every 6th second. It can be observed that the voltage signal at node g follows that of node a after exactly seven clock cycles. The logic level difference between node a and node g is only 0.1 V, and such low deterioration of the voltage levels ensures further logic propagation. Previous work on a two-bit shift register, based on lateral OECTs, suggests failure of the logic propagation in shift registers having six bits, or more, owing to logic level decays[44]. A video (Supplementary Movie 1) was recorded to show the signal '0' propagating along all the output nodes, from node g to node a. The propagation of the '0' input signal is shown on seven individual all-printed electrochromic displays[45], which further demonstrates

the compatibility between printed OECT-based logic circuits and external printed interface circuits.

Furthermore, we used the shift register to mimic the update of a printed seven-segment electrochromic display, which implies that for example the digit '4' can be shown on the display by providing seven clock pulses and the input pattern [0,1,1,0,0,1,1]. The voltage at node g (Q1) is presented in Fig. 5e when the above input pattern is shifted in, thus showing the correct logic levels throughout the whole-display update sequence. This was verified for all 10 digits [0–9]. By recording the logic voltage level at node g, the correct logic functionality of all 70 bits required for the display update sequence is demonstrated (Supplementary Figs. 16–20). Figure 5f shows the actual voltages that are measured when the input pattern [0,1,1,0,0,1,1] has been delivered to the parallel outputs interfacing with an external electronic circuit, in this case the pixel electrodes of all seven segments [a–g] of a printed electrochromic display. Segments exposed to a digital HIGH level at their respective output [b, c, f, g] are switched to ~– 1 V by the display driver circuitry (– 1 V is the display supply voltage that enables coloration of those segments), whereas the remaining segments [a, d, e] are kept at ~0 V, which corresponds to the digit '4'.

**Monolithic integration.** The robustness of the technology was further demonstrated by monolithic integration of OECT-based logic circuits and electrochromic displays, thereby forming an electronic system where screen printing solely is used as the deposition technique in the manufacturing process. The OECT architecture resembles the electrochromic display architecture; an electrolyte sandwiched between either the OECT channel and the gate electrode of the transistor, or between the pixel electrode and the counter electrode of the electrochromic display. Hence, the inclusion of a printed electrochromic display does not involve any additional processing steps. This concept was demonstrated by combining a seven-segment display with the BCD 4-to-7 decoder described earlier (Fig. 6). A video (Supplementary Movie 2) shows the full counting sequence [0–9] of the electrochromic display upon addressing the BCD 4-to-7 decoder with the corresponding binary input sequences [0000–1001]. Note that the colors of the segments might seem inverted, i.e., the ON-state is white and the OFF-state is blue, in Fig. 6 and Supplementary Movie 2, owing to omission of a blue-colored graphic shutter layer in the manufacturing process.

**Discussion**

The seven-bit shift register of the all-printed coplanar OECTs requires 114 transistors and 29 stages of logic propagation. This number of integrated OECTs is about one order of magnitude higher when compared with the number of transistors included in any other previous attempts to obtain (printed) OECT-based logic circuits. This achievement is attributed to a footprint reduction of the printed circuits by at least a factor of 12 when compared with previous reports. All individual components (OECTs, resistors, via interconnects and display segments) need to be functional in order to obtain a fully operational printed electronic system, hence, this also gives an indication of the robustness of the technology. The large-scale IC demonstrates the potential of all-printed OECT-based circuits for a great variety of applications, for example, in matrix-addressing, and (de)multiplexing of signals with increased number of rows and columns, also targeting counters, decoders, etc. In a hybrid electronics system, combining silicon-based electronics and printed electronics, the capability to implement complex functionalities of large-scale printed systems shares the workload of the silicon chips, and significantly reduces the functionalities and the number of pins required for the silicon chip,

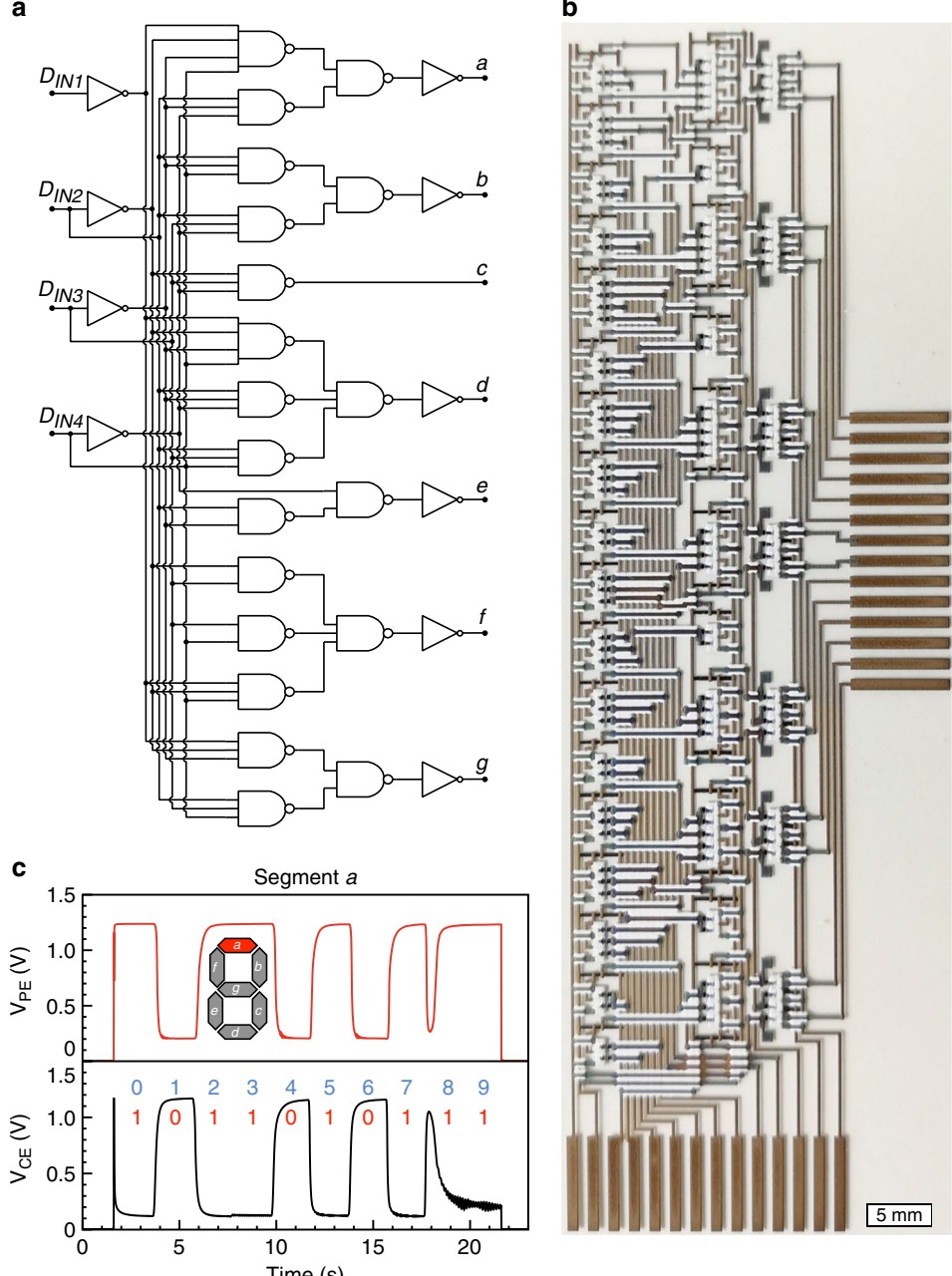

**Fig. 4 Binary-coded decimal 4-to-7 decoder. a** Schematics of a 4-to-7 decoder implemented with inverters and NAND gates, including two-, three-, and four-input NAND gates. **b** Photograph of the all-printed decoder that consists of 87 PEDOT:PSS-based OECTs (scale bar: 5 mm). Note that the details of the seven rightmost sub-circuits, representing the display driver circuitry including 28 additional OECTs, are omitted in this discussion. Four of the 14 pads located in the lower part of the image are input signals ($D_{IN1}$, $D_{IN2}$, $D_{IN3}$, $D_{IN4}$), whereas the other pads are dedicated to $V_{DD}$, $V_{SS}$, two GND pads, pixel voltage, enable signal, and four pads for testing. Each display segment (a, b, c, d, e, f, g) consists of one pixel electrode and one counter electrode, which results in seven pairs of output pads located on the right side of the image. These output pads are used for circuit characterization, or to connect the circuit with an external display device. **c** Logic voltage output sequence ($V_{PE}$, data shown in red are the logic level controlling the display pixel electrode) of segment a [1,0,1,1,0,1,0,1,1,1] when the input signal corresponds to the digits [0,1,2,3,4,5,6,7,8,9]. The data shown in black are the logic voltage level controlling the display counter electrode ($V_{CE}$), hence, the two data sets are inverse to each other. The input signal representing the counting sequence from 0 to 9 is provided by changing the input signals $D_{IN1}$ to $D_{IN4}$, e.g. [0,0,0,0], [0,0,0,1], [0,0,1,0], etc., every two seconds. The glitch at ~18 s is owing to the fact that the switching time varies between different OECTs within the circuit

thus greatly improving cost efficiency. For instance, with an 8-pin microcontroller that costs < 1 US$, it is then possible to drive all-printed multidigit seven-segment displays without losing the ability of sensor signal acquisition and processing. The ability to print low-voltage large-scale circuits on flexible substrates also enables many IoT applications, such as smart sensor labels, tags, sticks, signages, etc., on various substrates, e.g., conformable substrates, paper, and cloth. In combination with an array of sensors, a wearable device based on the presented technology could be developed for further diagnostic purposes, e.g. to monitor temperature, blood oxygen, or health status parameters expressed in sweat, on human skins.

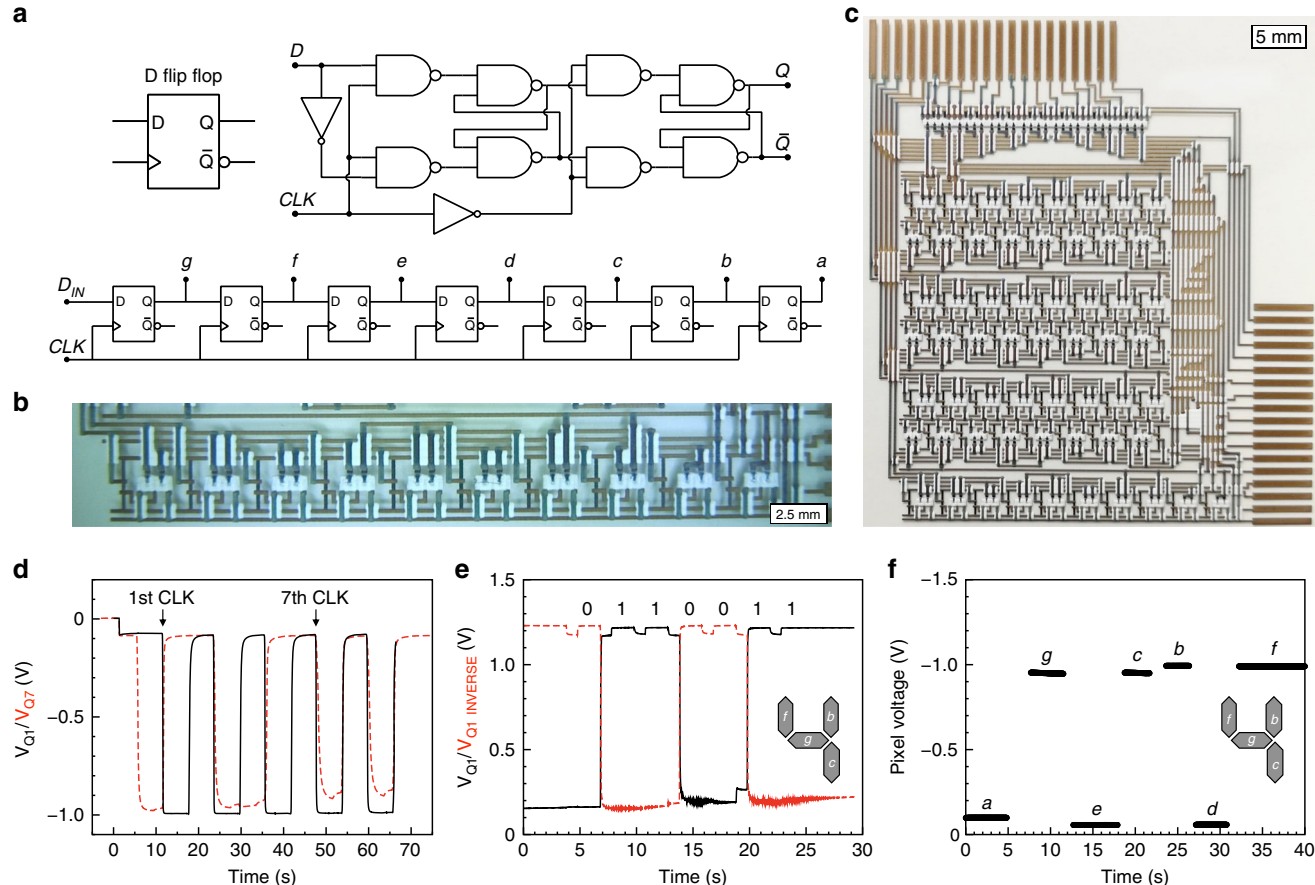

**Fig. 5** Seven-bit shift register based on organic electrochemical transistors. **a** Schematic and symbol of a Master-Slave D flip-flop implemented via inverters and two-input NAND gates, and a seven-bit shift register cascading seven stages of D flip-flops. **b**, **c** Photographs of the all-printed D flip-flop (**b**, scale bar: 2.5 mm) and the seven-bit shift register (**c**, scale bar: 5 mm). **d** Voltage outputs from the shift register at display segments g and a, corresponding to the first (Q1) and the seventh (Q7) stage/bit, respectively. The circuit is triggered by a clock pulse every 6th second. **e** Voltage outputs from the shift register at node g (Q1) and its complementary Q1$_{INVERSE}$ when the input pattern [0,1,1,0,0,1,1] is shifted in by seven clock cycles to display the digit '4'. **f** The voltage levels of the pixel electrodes of all seven segments (a, b, c, d, e, f, g) when the display shows the digit '4'. The parallel output voltages shown in **f** were obtained by subsequent measurements of individual display segments (each recorded during ~5 s) during a total time of ~40 s, hence, this test also demonstrates the non-volatility of the shift register circuit. A total retention and operational performance for the circuit of more than several minutes have been demonstrated

In summary, we have developed and manufactured, by using only screen printing processing, all-printed large-scale integrated circuits (4-to-7 decoder and seven-bit shift register) from coplanar top-gate PEDOT:PSS-based OECTs. The OECTs exhibit relatively fast switching, small footprint, excellent uniformity and a simple manufacturing route. Logic gates, including inverters and NAND gates, were investigated as the building block of the large-scale circuits. We demonstrated that both the 4-to-7 decoder and the seven-bit shift register deliver correct logic functionalities. In addition, the capability to drive all-printed electrochromic displays is shown, either with standalone or monolithically integrated electrochromic displays. The technology of all-printed large-scale ICs on flexible substrates, operated at low voltages, offers an ultra-low-cost solution for many IoT applications.

## Methods

**Materials**. The plastic substrate of polyethylene terephthalate (PET) *Polifoil Bias* is purchased from *Policrom Screen*. PEDOT:PSS *Clevios SV3*, purchased from *Heraeus*, serves as the electrochemically active transistor channel and gate electrodes, as well as the pixel and counter electrodes of the electrochromic displays. The electrolyte *AFI VV009*, which is based on poly(diallyldimethylammonium chloride) dissolved in water, solid particles providing opacity, initiator molecules and binder molecules capable of forming a cross-linked network through radical polymerization upon UV light irradiation, is provided by RISE Acreo on commercial

terms. The screen printable and UV-curable electrolyte is used for both OECTs and electrochromic displays. Carbon ink *7102* and *7082* printing pastes (purchased from *DuPont*) are used for drain/source contacts (to minimize the reduction front) and printed resistors, respectively. To obtain printed resistors with approximately five times higher sheet resistance, and thereby lower the footprint of the printed resistors, the *7082* paste was diluted by adding ~20% of the *5036* paste from *DuPont* (Supplementary Fig. 7). Silver ink (*Ag 5000* purchased from *DuPont*) and insulating ink (*5018* purchased from *DuPont*) are used for printed interconnects and isolation, respectively.

**Manufacturing of OECTs, circuits, and electrochromic displays**. All materials were deposited by flatbed sheet-fed screen printing equipment (*DEK Horizon 03iX*) on top of PET plastic substrates, the printed features of the design layout cover approximately the area of an A4 sheet. The screen printing tools, *V-screens* provided by *PVF*, were based on polyarylate threads. The *DEK Horizon 03iX*, which has a process alignment capability of ±25 μm, in combination with the high-quality screen printing tools ensure high manufacturing yield by that uniform alignment accuracy across the whole substrate area is obtained; misalignment or ink smearing by > ~50 μm will inevitably result in malfunctioning devices and circuits. *Ag 5000* silver ink is first deposited on the substrate as contact pads and interconnect wires. A layer of PEDOT:PSS is then patterned between the silver contact pads to form the transistor channel, followed by the carbon electrodes, resulting in an active channel area of $100 \times 150$ μm$^2$. The PEDOT:PSS layer is simultaneously deposited to form the display pixel electrode layer. The silver, the PEDOT:PSS and the carbon inks are all dried at 120 °C for 5 min. An insulating layer of *5018* ink is then deposited and UV-cured to define the area of the subsequently deposited gate electrode. The electrolyte required to enable the display functionality is deposited in the same print step. Another layer of PEDOT:PSS is deposited on the electrolyte

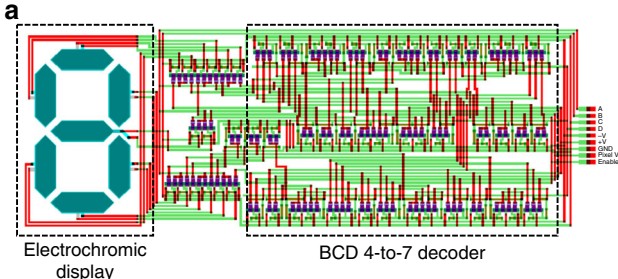

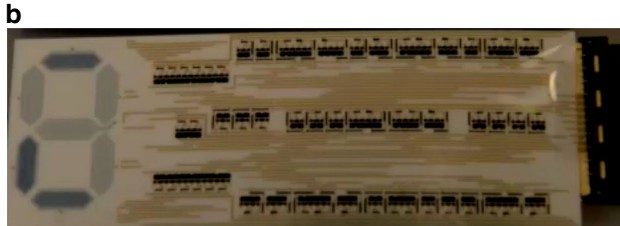

**Fig. 6** Monolithic integration forming an all-printed electronic system. **a** Layout illustrating monolithic integration of the electrochromic seven-segment display and the BCD 4-to-7 decoder. The OECTs located between the BCD 4-to-7 decoder and the electrochromic display are used to drive the display segments upon receiving the correct logic voltage levels from the decoder circuit. **b** Optical image of the electrochromic display and decoder monolithically integrated on a flexible substrate by only using screen printing in the manufacturing process. The digit '4' is visible on the electrochromic display after addressing the 4-to-7 decoder with the binary input sequence [0,1,0,0]. The total area of the all-printed electronic system is ~40 cm², i.e., smaller than a credit card

patterns as the gate electrodes and the display counter electrodes. The resistors in the circuits are formed by the carbon ink *7082*. The crossings of interconnects are created by the deposition of the insulating *5018* ink followed by the deposition of *Ag 5000*, the latter also forms the conductors of the display device.

**Characterization of OECTs and circuits**. All measurements are performed in a controlled environment at a temperature of ~20 °C and at a relative humidity of ~45% RH. Transfer and dynamic switch measurements were performed by using a semiconductor parameter analyser (*HP/Agilent 4155B*) and a function generator (*Agilent 33120A*). The printed circuits were characterized by using a data acquisition card (*DAQ card PCI-6723* from *National Instruments*). The voltage supply [$V_{DD}$, $V_{SS}$] for the logic gates, 4-to-7 decoders, and seven-bit shift registers is [+5 V, −5 V], whereas the voltage supply for the electrochromic displays was − 1 V (Fig. 5f) or − 1.5 V (Fig. 6 and Supplementary Movies 1 and 2). The input signals controlling the printed circuits were also supplied by the DAQ card, and the frequency was varied for the different circuits being characterized. One hertz was used for standalone OECTs, inverters and NAND gates, whereas lower frequencies were used for the more complex 4-to-7 decoder (digits updated every two seconds) and seven-bit shift register circuits (new clock pulse every 6th second) to provide sufficient time to ensure complete propagation of the input signals through all OECTs within the circuits. Higher switching frequencies could possibly have been used, especially for the seven-bit shift register circuit, but no attempts to determine the frequency limits of the circuits have been conducted.

## Data availability

The data that support the findings of this study are available from the corresponding author upon request.

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

## Acknowledgements

This project was financially supported by the Swedish Foundation for Strategic Research (SE13–0045 and RIT15-0119), the Knut and Alice Wallenberg Foundation (2012.0302), the Önnesjöstiftelsen, VINNOVA and the Swedish Research Council (2016-03979). We thank DI Gerald Holweg, Pre Development Manager at Infineon Technologies Austria AG, for valuable discussions.

## Author contributions

P.A.E., D.T., R.F., S.F., G.G., and M.B. conceived and designed the project. D.T., V.K., and R.F. were responsible for circuit designs and simulations. J.S., with assistance from P.A.E. and R.L., was responsible for designing the screen printing tools and the screen printing manufacturing process. P.A.E. and R.L. performed electrical characterization on printed OECTs and OECTs-based circuits, as well as demonstration of monolithically integrated OECTs and electrochromic displays manufactured by screen printing. P.A.E., R.L., D.T., S.F., M.B. wrote the paper. All authors contributed to discussions and manuscript preparation.

## Competing interests

Acreo AB filed a patent application related to the electrolyte used in this work, through the World Intellectual Property Organization, international publication number WO 2012/136781, filed on 5 April 2012, granted on 19 June 2018 with patent number US 10001690B2. P.A.E. and M.B. filed a patent application via Acreo AB, through the World Intellectual Property Organization, international publication number WO 2011/042430, filed on 5 October 2010, granted on 19 August 2014 with patent number US 8810888B2. All other authors declare no competing interests.
