## [Peer Review File · Nature Communications]

Reviewers' comments:

Reviewer #1 (Remarks to the Author):

Summary: The work presents a significant increase in the sophistication of integrated circuits formed from OECTs. It builds on previous device and circuit integration work by the Linköping group but is a substantial advance on the previous results. The work is well documented and the conclusions solid.

1. There are a few other groups that have explored OECT integration, e.g. Stadlober's. While their work is cited (ref 34), it is as an aside in a discussion of results. It would be better to comprehensively summarize prior work in this field to put this advance in appropriate context. I would be particularly interested to see how the number of transistors in an operating circuit has evolved over last decade or so that this has been pursued, however this is up to the authors discretion.
2. The paper present impressive results of large-scale functional circuits and demonstrates several critical features that make this possible such as logic level restoration and state retention. However, there is almost no discussion of what advances (e.g. over reference 34 or the authors' previous work) enable these capabilities. That is, there is no "why", only "what" was shown.
 - a. For example, on page 4 the authors mention that NAND/NOT logic tends to suffer from logic level decay but then use this logic and show that they do not suffer this problem. This begs explanation.
 - b. The 40 s retention is a good demonstration. A supplementary figure showing a sample and hold measurement would be stronger. What is the retention limit of OECTs in the circuit? Does it depend on fanout or other circuit topology?
3. In figure 4c between 17 and 18 seconds there is a positive voltage swing on VCE. I presume this is due to other gates in the circuit not all switching simultaneously. Figure S5 shows some similar behavior. Please explain. You may wish to include some form of tick marks showing the clock to indicate when logic levels are sampled and thus must be stable.
4. The switching time of the OECTs is quoted as 20 to 30 ms depending on direction. Figure 3 operates at about 1 second per logic level. Figure 4 is about 2 seconds. Figure 5d is about 6 seconds. How did the authors choose the clock period? The results are quite good and I do not mean to say faster speed is needed, it simply is the case that the transistor speed and circuit speed are quite different and this should be explained.
5. Similar to the last two comments, the settle time in Figure S8 is clearly much longer than 10s of ms. Please explain.
6. I was unable to find any details of (e.g. the chemical nature of) the printed electrolyte on the RISE website. Please provide a description or generally accessible reference to this material. I note that one of the authors is from the company and thus am concerned that this material may not be fully available to others wishing to repeat the authors' work. I would find this unacceptable as a reviewer.
7. In figure 2c and 3c, please include arrows or some notation that indicate the direction of the voltage sweep.

Reviewer #2 (Remarks to the Author):

The manuscript, entitled "All-Printed Large-Scale Integrated Circuits Based on Organic Electrochemical Transistors" by Peter A. Ersman et al., presents large-scale screen printing fabrication of an OECT digital circuit. The authors insist that the printed OECT sequential logic circuits can reduce silicon chip pad area, thereby achieving more efficient ASIC design. This paper demonstrates BCD 4-to-7 decoders and 7-bit shift registers, operating at less than 1.5 V including over 100 OECTs. However, the reviewer concludes that the works shows incremental engineering developments from their previous studies on C-C OECTs (Ersman et al. Flex. Print. Electron. 2017)

and cannot find a significant scientific novelty in this manuscript, In addition, the main motivation of this work, which is that 'printed OECT circuits can reduce silicon chip pad area', seems to be generally applied to any types of printed circuits, which compromise the novelty of the manuscript. Therefore, although the results are well presented, the lack of scientific advance/novelty makes this paper more suitable for a more specialized journal. For the better quality of the manuscript, the reviewer suggests that the authors need to (i) make the abstract/introduction more concise and attractive, (ii) discuss major advances of this work from their previous papers, (iii) emphasize the monolithic integration part because the all-printed monolithic integration of circuits/displays has been rarely reported, (iv) measure gate delay more rigorously by giving the load condition (ohm) or fabricating a OECT ring oscillator.

Reviewer #3 (Remarks to the Author):

This paper describes the fabrication of screen-printed OECT logic with a relatively high level of complexity, and its interconnection with a monolithically integrated 7-segment display. It is impressive from the point of view of engineering that the process employed was robust enough to manufacture hundreds of functional devices. The topic is also of general interest and involves different disciplines. However, I have problems understanding what is the key discovery or achievement compared to the state of the art in references 26-32. Since other authors have already printed similar stand-alone OECT with the same geometry and materials, and have also fabricated logic circuits (although less complex) using OECT, this paper seems to introduce only an incremental advancement in the field rather than a breakthrough.

Other than this major comment, other minor comments to improve the clarity of the paper are the following:

- Abstract: It is not clear what "interface devices" mean.
- Fig 1a. I would suggest to draw a larger ASIC chip on the left side (many contact pads) than on the right side. This would help visualization of the concept described in the paper of minimizing Si surface area.
- Figure 1b. How is the estimation of cost made? There is not reference or further explanation.
- Page 5. An explanation (quantified) for the selection of the values for the resistors ladder is missing. Which kind of calculations were made to optimize the selected values?
- Figure 4c. Consider to add a sketch of a 7-segment display labeling each segment, for readers non-familiar with this kind of displays.
- The combination of shift register, BCD and display has not been shown. It would add extra value to the paper to see the whole system working together, although I understand that this may introduces a considerable amount of extra work. Also, showing the interconnection of the printed circuits with a Si chip with only 2 pins would demonstrate the advantages of the concept claimed in the introduction.
- Methods / Manufacturing: Could you justify why PEDOT:PSS is used as gate electrode instead of Carbon or Ag? Likewise, why is C used as Source and Drain material instead of Ag or PEDOT:PSS?
- Methods, manufacturing: How is the dielectric used to define the electrolytic gate?

Answers to the reviewer comments

Reviewer #1 (Remarks to the Author):

Summary: The work presents a significant increase in the sophistication of integrated circuits formed from OECTs. It builds on previous device and circuit integration work by the Linköping group but is a substantial advance on the previous results. The work is well documented and the conclusions solid.

1. There are a few other groups that have explored OECT integration, e.g. Stadlober's. While their work is cited (ref 34), it is as an aside in a discussion of results. It would be better to comprehensively summarize prior work in this field to put this advance in appropriate context. I would be particularly interested to see how the number of transistors in an operating circuit has evolved over last decade or so that this has been pursued, however this is up to the authors discretion.

The results of the revised version include a detailed comparison with Refs. 32 and 34 (now Refs. 42 and 44 of the revised manuscript, respectively), especially with respect to the number of printed OECTs in the circuits and the impact of OECT footprint reduction.

2. The paper present impressive results of large-scale functional circuits and demonstrates several critical features that make this possible such as logic level restoration and state retention. However, there is almost no discussion of what advances (e.g. over reference 34 or the authors' previous work) enable these capabilities. That is, there is no "why", only "what" was shown.

Footprint reduction is the key to obtain large-scale functional circuits, as here reported. The footprint reduction not only makes it possible to print complex OECT-based circuitry on a reasonably small area, it also enables an improvement of the manufacturing yield, a necessary feature to achieve circuits including more than 100 functional OECTs. The maturity of the technology presented in Refs. 32 and 34 (now Refs. 42 and 44 of the revised manuscript, respectively) would have been insufficient in terms of total area and size of the resulting circuitry, and manufacturing yield. Regarding detailed discussion on the advancement over prior achievement; several new sections have been added to the manuscript, especially in the abstract, page 1, and in the "Logic gates based on printed OECTs" section on page 6.

a. For example, on page 4 the authors mention that NAND/NOT logic tends to suffer from logic level decay but then use this logic and show that they do not suffer this problem. This begs explanation.

We thank the reviewer for pinpointing this unclear statement. Indeed, this limited sentence on page 4 appears to be confusing, and the paragraph is now extended to provide a sufficient description of the solution to achieve digital signal propagation, which in part includes an increase in driving voltage.

b. The 40 s retention is a good demonstration. A supplementary figure showing a sample and hold measurement would be stronger. What is the retention limit of OECTs in the circuit? Does it depend on fanout or other circuit topology?

The results of Figure 5f is based on a sequential measurement that simply took 40 s in total. The (manual) recording of each output terminal is here around 5 s. An additional sentence in the caption of Figure 5f was also included stating: "A total retention and operational performance for the circuit of more than several minutes has been demonstrated."

3. In figure 4c between 17 and 18 seconds there is a positive voltage swing on VCE. I presume this is due to other gates in the circuit not all switching simultaneously. Figure S5 shows some similar

behavior. Please explain. You may wish to include some form of tick marks showing the clock to indicate when logic levels are sampled and thus must be stable.

This is a very good comment. Yes, indeed this is due to differences in switching time between OECTs. And, at the same time there is a corresponding negative voltage swing on V_{PE} , which is natural since they should behave oppositely to each other. Figure 4c and Figures S10-S15 refer to the 4-7 decoder, so there is no clock signal in those particular measurements. Instead, a logic combination is given as an input, in order to count from 0 to 9, i.e. [0,0,0,0], [0,0,0,1], [0,0,1,0], and so on. Each input signal lasts for two seconds. The voltage swings are caused by that at least one OECT device is switching slightly slower, but the logic system eventually wiggles into the correct logic state. This has now been clarified in the caption of Figure 4c.

4. The switching time of the OECTs is quoted as 20 to 30 ms depending on direction. Figure 3 operates at about 1 second per logic level. Figure 4 is about 2 seconds. Figure 5d is about 6 seconds. How did the authors choose the clock period? The results are quite good and I do not mean to say faster speed is needed, it simply is the case that the transistor speed and circuit speed are quite different and this should be explained.

Again, excellent comment. Indeed, the switching time of each OECT is in the range 20-30 ms. But, for the complete circuits, we wish to ensure that the signals given at the input has sufficient amount of time to propagate through all OECTs of the circuits. The purpose of the measurements, presented in Fig. 4 and Fig. 5, is to demonstrate *counting* from 0 to 9 in both the decoder circuit and in the shift register circuit, and to record this by measuring the voltage levels at the output terminals. Therefore, we chose a slow frequency on the input side to ensure complete signal propagation and stability. Each circuit contains approximately 100 OECTs, and their collective behavior finally results in the output voltage levels. A more extensive description of the addressing protocols and a motivation for the chosen frequency is now included on page 14 of the revised manuscript (Method section, Characterization of OECTs and Circuits).

5. Similar to the last two comments, the settle time in Figure S8 is clearly much longer than 10s of ms. Please explain.

The answer given to comment “4” is valid also here. The switching time of each OECT is 20-30 ms also for the OECTs presented in Fig. S8 (now Fig. S13 of the revised manuscript) and the time period to achieve complete signal propagation is set to a relatively much longer time.

6. I was unable to find any details of (e.g. the chemical nature of) the printed electrolyte on the RISE website. Please provide a description or generally accessible reference to this material. I note that one of the authors is from the company and thus am concerned that this material may not be fully available to others wishing to repeat the authors' work. I would find this unacceptable as a reviewer.

We thank the reviewer for highlighting the missing information. The electrolyte printing ink is based on poly(diallyldimethylammonium chloride), containing solid particles for opacity, initiator molecules and binder molecules capable of forming a cross-linked network through radical polymerization upon UV light irradiation. This water-based electrolyte is commercially available through RISE Acreo and the available details about its content are given in the Method section (Materials, page 13).

7. In figure 2c and 3c, please include arrows or some notation that indicate the direction of the voltage sweep.

Arrows are now included in Figures 2c and 3c.

Reviewer #2 (Remarks to the Author):

The manuscript, entitled “All-Printed Large-Scale Integrated Circuits Based on Organic Electrochemical Transistors” by Peter A. Ersman et al., presents large-scale screen printing fabrication of an OECT digital circuit. The authors insist that the printed OECT sequential logic circuits can reduce silicon chip pad area, thereby achieving more efficient ASIC design. This paper demonstrates BCD 4-to-7 decoders and 7-bit shift registers, operating at less than 1.5 V including over 100 OECTs. However, the reviewer concludes that the work shows incremental engineering developments from their previous studies on C-C OECTs (Ersman et al. Flex. Print. Electron. 2017) and cannot find a significant scientific novelty in this manuscript,

Several scientific and technological developments have here been carried out, including new ink formulations, printing processes, layout and biasing protocols. The report therefore includes a collection of several advancements, and not only one single scientific leap! Together, these advancements allow us to produce LSI circuits entirely based on OECTs, outperforming previous achievements with at least an order of magnitude with respect to complexity and size. The truly unique achievement here reported, relates to the fact that we can manufacture OECT-LSIs with excellent digital signal propagation. And with those at hand, we can integrate Si-chip technology with printed electronics in a manner never reported before, twinning Si chips with OECT-based IoT labels. It is our opinion, that this is of interest for the readers of Nature Communications. Improved motivations and better clarifications of our advancements are found on pages 5-7 and in the abstract of the revised manuscript.

In addition, the main motivation of this work, which is that ‘printed OECT circuits can reduce silicon chip pad area’, seems to be generally applied to any types of printed circuits, which compromise the novelty of the manuscript. Therefore, although the results are well presented, the lack of scientific advance/novelty makes this paper more suitable for a more specialized journal.

There might be other reports on the topic of printed circuits aiming at reducing the pin count on the Si chips, but there are no such reports targeting an all-printed circuit approach, with complete integration of LSI/Output device that operates at compatible driving protocols. Moreover, our manuscript reports an all-printed LSI approach in which screen printing has been used as the only deposition/ patterning method, which gives us a very robust production protocol. Several other reports claim “printed” or “all-printed” as the theme for the work carried out. However, typically, these demonstrate transistors and circuits where only one, or a few, layers are deposited by printing techniques, while the remaining layers are deposited by other means, typically using spin-coating or vacuum deposition. Integration, or true *printegration*, of high complexity, here based on all-printed OECT-LSIs including more than 100 OECTs and electrochromic displays, are demonstrated for the first time herein. Previous reports have demonstrated very much less complex circuits with only few OECTs, but still requiring larger substrate area. Also, the pin count on the Si chip is reduced from 7 to 4 in the 4-7 decoder, and from 7 to 2 in the 7-bit shift register.

For the better quality of the manuscript, the reviewer suggests that the authors need to

(i) make the abstract/introduction more concise and attractive

We revised the abstract and introduction, highlighting the motivations and the major advancements.

(ii) discuss major advances of this work from their previous papers

The major advances of this work have already been motivated in this response letter, and it has also been added to pages 5-7 of the revised manuscript.

(iii) emphasize the monolithic integration part because the all-printed monolithic integration of circuits/displays has been rarely reported

The all-printed monolithic integration of OECT-based logic circuits and the electrochromic displays was already highlighted as a separate section and the motivation for this work have been stressed even further in the abstract and introduction.

(iv) measure gate delay more rigorously by giving the load condition (ohm) or fabricating a OECT ring oscillator.

This is a good comment, but a ring oscillator only contains e.g. 5 or 7 OECTs, while the large circuits presented herein contain more than 100 OECTs. And since they are manufactured through relatively rough screen printing methods, they are likely to behave somewhat different. If only one OECT behaves differently, e.g. if it shows longer switching time, the propagation of the logic signal will be delayed just because of this OECT device, and in these large circuits it is impossible to determine which particular OECT that is behaving differently. However, the propagation delay in each stage of a ring oscillator has now been evaluated, and the results have been added on page 8 of the revised manuscript.

Reviewer #3 (Remarks to the Author):

This paper describes the fabrication of screen-printed OECT logic with a relatively high level of complexity, and its interconnection with a monolithically integrated 7-segment display. It is impressive from the point of view of engineering that the process employed was robust enough to manufacture hundreds of functional devices. The topic is also of general interest and involves different disciplines. However, I have problems understanding what is the key discovery or achievement compared to the state of the art in references 26-32. Since other authors have already printed similar stand-alone OECT with the same geometry and materials, and have also fabricated logic circuits (although less complex) using OECT, this paper seems to introduce only an incremental advancement in the field rather than a breakthrough.

It is true that there are previous reports presenting printed OECTs and OECT-based circuits of less complexity. But as already mentioned in this response letter, one should keep in mind that the circuits presented herein would not have been possible to print a few years ago. A number of print iterations, different designs, ink development and process developments have matured the technology to finally reach the results presented herein. As also mentioned in a comment to reviewer #1, footprint reduction has been the key to obtain the large-scale functional circuits of the manuscript. The footprint reduction not only makes it possible to print this kind of complex OECT-based circuitry on a reasonably small area, to make the circuit compatible with e.g. electronic smart labels and printed electrochromic displays, but the footprint reduction has also enabled an improvement of the manufacturing yield of the transistor devices that is required to make circuits having more than 100 functional OECTs. The maturity of the technology presented in Refs. 32 and 34 (now Refs. 42 and 44 of the revised manuscript, respectively) would have been insufficient in terms of total area of the resulting circuitry, and the manufacturing yield that was demonstrated in Ref 42 would not have been sufficient to establish the printed circuits of this manuscript.

Other than this major comment, other minor comments to improve the clarity of the paper are the following:

- Abstract: It is not clear what “interface devices” mean.

Interface devices can be either on the input side (e.g. a keyboard) or on the output side (e.g. a display), and denote some kind of circuit or device through which user interaction can take place. This was not obvious in the manuscript and has now been clarified.

- Fig 1a. I would suggest to draw a larger ASIC chip on the left side (many contact pads) than on the right side. This would help visualization of the concept described in the paper of minimizing Si surface area.

This is a very good suggestion. Figure 1a has been changed according to the reviewer’s comments.

- Figure 1b. How is the estimation of cost made? There is not reference or further explanation.

We assumed using an 8-inch silicon wafer having a total production cost of 1500 US\$. However, it is true that some important information is missing, e.g. the assumed pad length, width and pitch of the silicon chip. This, in turn, results in a certain chip area for a certain number of pads, as well as it gives the total number of chips that can be obtained from each wafer. This is how the curves in Figure 1b are obtained, and this has been clarified in the caption of Figure 1b.

- Page 5. An explanation (quantified) for the selection of the values for the resistors ladder is missing. Which kind of calculations were made to optimize the selected values?

First of all, we observed the ON and OFF current levels of the OECT device. The resistance in the ON-state is about 1 k Ω , and we wanted the value of R3 to be at least 20 times higher in order to ensure proper output voltage level. At the same time, it is desired to maintain relatively low resistor values, since this shortens the switching time of the logic circuits. We also made adjustment to the supply voltage, from +/-3 V (Ref. 32, now Ref. 42 in the revised manuscript) to +/-5 V in this manuscript, to minimize the voltage strain at the drain electrode. The evolution of parasitic current levels vs. the applied drain voltage for the OECT OFF-state has now been included in Fig. S9, since this is an important design parameter. These results, combined with the simulation results now included in Figs. S5 and S6, provide guidelines on how to choose the resistor values for the targeted supply voltage and logic output voltage levels. Besides the desire to optimize the switching performance of the circuits, the choice of resistor values is also crucial to minimize the voltage strain at the drain electrode, and thereby prolong the operational lifetime of the OECT devices. This information is now included at page 5 of the revised manuscript.

- Figure 4c. Consider to add a sketch of a 7-segment display labeling each segment, for readers non-familiar with this kind of displays.

As suggested by the reviewer, we added a sketch of a 7-segment display to Figure 4c.

- The combination of shift register, BCD and display has not been shown. It would add extra value to the paper to see the whole system working together, although I understand that this may introduces a considerable amount of extra work. Also, showing the interconnection of the printed circuits with a Si chip with only 2 pins would demonstrate the advantages of the concept claimed in the introduction.

The 4-7 BCD decoder is a logic circuit that is commonly used to convert four addressing signals at the input to seven output signals, to operate a 7-segment display. Hence, the number of addressing signals

can be reduced from seven to four, as compared to when using direct addressing of the display. This fully working screen printed system, including also the monolithically integrated display, is demonstrated in Figure 6 of the manuscript and in Video V2 in the Supporting Information. To further reduce the number of required I/Os, we also developed the screen printed 7-bit shift register, which is another way of addressing a 7-segment display. The shift register relies on a serial input signal and seven parallel output signals, but this adds the requirement of also having a clock signal in the circuit. Even though the circuit was designed as a 7-bit shift register in this particular work, it could consist of an arbitrary number of output signals, but still only requiring two input signals; one data signal and one clock signal. This complete system is capable of updating seven display pixels, in this case screen printed but not monolithically integrated, by only using the two input signals of the shift register, as demonstrated in Video V1 in the Supporting Information.

We agree that it would have been an even better demonstrator in case of using a silicon chip with only two pins, but from a characterization point of view, it was very convenient to use a DAQ-card while testing the circuits, since this enable full freedom in terms of supply voltage and frequency of the input signals. In addition to this, pick and place assembly of microcontrollers on flexible substrates has been demonstrated before, in other smart label applications, and would not add any extra novelty to this manuscript.

- Methods / Manufacturing: Could you justify why PEDOT:PSS is used as gate electrode instead of Carbon or Ag? Likewise, why is C used as Source and Drain material instead of Ag or PEDOT:PSS?

Silver as the gate electrode will not work, except for a few initial switches. Ag is electrochemically active, and continuous cycling will cause Ag ions to enter into the electrolyte and thereby negatively affect the OECT switching performance. Carbon works as the gate electrode, but it shows limited charge capacity, hence, the switching voltage increases from about 1.2-1.3 V in the case of a PEDOT:PSS-based gate electrode to >2 V in the case of a carbon-based gate electrode. Such high gate voltage would imply an increased voltage strain between the gate and drain electrodes, which in turn implies shorter operational lifetime.

The reason for using carbon as the drain and source electrodes is that it minimizes the reduction front, thus improving the OECT response time. Ag-based source and drain electrodes would not work for the same reason as mentioned above regarding using Ag as the gate electrode, it is simply too active from an electrochemical point of view. PEDOT:PSS would clearly work as the source and drain electrodes, in fact, it even gives better OECT performance in terms of on/off-ratio and operational lifetime. However, the so called reduction front severely limits, or even completely disables, the switching frequency of OECTs and OECT-based circuits relying on PEDOT:PSS as the source and drain electrodes.

- Methods, manufacturing: How is the dielectric used to define the electrolytic gate?

The OECTs could, in fact, be manufactured without this insulating layer. The reason for printing the insulating layer is simply to create a well-defined area for the subsequently printed electrolyte layer. The insulating layer ensures that the electrolyte only comes into contact with the PEDOT:PSS-based OECT channel and minimized areas of the source and drain carbon electrodes. Without the insulating layer, the areas of the electrolyte/carbon interfaces would increase, which in turn would result in increased parasitic current levels, i.e. higher off-current levels.

REVIEWERS' COMMENTS:

Reviewer #1 (Remarks to the Author):

The authors have fully addressed my concerns by making appropriate changes to the manuscript.

Reviewer #2 (Remarks to the Author):

In this revised manuscript, authors have addressed all the points raised by the reviewer. Therefore, this manuscript is now ready for publication.

Reviewer #3 (Remarks to the Author):

Second review

General comment:

I appreciate the comments by the authors and I understand that different milestones in material and process development have contributed during the last years to make possible the complexity demonstrated in this paper. However, the authors do not clearly indicate if any of these important enabling milestones has been developed within this work. It seems that this work reports mostly on an improved demonstration of the technology previously developed. Similar comments apply to the reduction of footprint. The device miniaturization (which could be seen as a breakthrough) has been enabled mostly by a better screen mesh, but not by any specific technique reported in this paper other than the reformulation of the ink used for the resistor.

BCD integration comment:

Why the circuit in Fig. 4b is so different from the circuit in Fig. 6 if they are both 4-7 BCD decoders? In those figures, what are the inputs and what the outputs? In Fig. 4b, there are 2x14 pads, which I assume are the outputs duplicated. Where are the inputs? This info should be in the figure. In Fig. 6, it is hard to spot inputs and outputs as well.

The authors did not follow up my comment regarding the combination of shift register, BCD and display. I understand the role of each circuit, but it would have added value to see a monolithically integration of the 4-7 BCD (Fig. 4b) and the shift register (Fig. 5c) with the BCD.

I understand the justification not to use a Si chip in the demonstration.

Methods / Manufacturing for contact materials comment:

The authors should include this explanation with references to the reduction front issue in the paper.

Methods / Manufacturing for dielectric patterning comment:

I thank the authors for the clarification. The explanation is clear but it does not appear in the text, neither is this layer shown in the cross-section of the device.

From the text sentence "An insulating layer of 5018 ink is then deposited and UV-cured to define the area of the subsequently deposited gate electrolyte" the reader cannot know if the dielectric is patterned on the gate or around it (which is the case according the extra explanation provided by the authors).

Other small comments:

Abstract:

- It is not clear what the authors mean with the "former" and the "latter" devices.
- The sentence: "thus enabling few Si-output terminals to drive monolithically integrated all-

printed 7-segment electrochromic displays" is misleading since it looks like the authors have demonstrated this interconnection

- "driving protocols" -> what are those in the paper?

Page 4

- "This was obtained by increasing the margins of the voltage output levels" -> what from which value?

- "the minimized voltage strain implies..."-> minimizes from which value? Justify the selection of +/- 5 V

Page 6

- Explain the following in Fig. S6: "Additionally, the chosen values imply that the resistor ladder is tolerant to ~15% of resistance variation of the all-printed resistors (Fig. S6)"

ANSWERS TO REVIEWER #3's COMMENTS

General comment:

I appreciate the comments by the authors and I understand that different milestones in material and process development have contributed during the last years to make possible the complexity demonstrated in this paper. However, the authors do not clearly indicate if any of these important enabling milestones has been developed within this work. It seems that this work reports mostly on an improved demonstration of the technology previously developed.

Similar comments apply to the reduction of footprint. The device miniaturization (which could be seen as a breakthrough) has been enabled mostly by a better screen mesh, but not by any specific technique reported in this paper other than the reformulation of the ink used for the resistor.

This work comprises a collection of several scientific and technological advancements, including new ink formulations, printing processes, layout and biasing protocols. All together, these developments allowed us to produce LSI circuits entirely based on OECTs, outperforming previous achievements with at least an order of magnitude with respect to complexity and size. The footprint reduction is the key parameter of this contribution, since it not only allows screen printing of OECT-based circuits on an adequate area, but also provides higher manufacturing yield, and this has been obtained by improving e.g. the resistor ink and the screen printing tools reported within this work. The aim of this study was mainly to demonstrate 1) how the number of input signals required from the addressing electronics can be minimized by taking advantage of printed OECT-based circuits, and 2) how a 7-segment display can be updated by utilizing printed circuitry having an area comparable to the display area. Correct signal propagation through the relatively large number of logic gates reported in these circuits would not have been possible without careful circuit simulations and sub-sequent circuit evaluations. The circuit simulations have provided guidelines related to the choice of resistor values (i.e., resistor ink optimization), supply voltage as well as targeted logic output voltage levels. Hence, the work being presented in this manuscript is the result of a holistic approach on taking the OECT technology to the next level of sophistication.

BCD integration comment:

Why the circuit in Fig. 4b is so different from the circuit in Fig. 6 if they are both 4-7 BCD decoders? In those figures, what are the inputs and what the outputs? In Fig. 4b, there are 2x14 pads, which I assume are the outputs duplicated. Where are the inputs? This info should be in the figure. In Fig. 6, it is hard to spot inputs and outputs as well.

The reason for the different circuit layout is simply explained by that the circuit demonstrated in Fig. 6 contains no output pads at all. The logic outputs of the decoder in Fig. 6 are directly connected with the monolithically integrated display, hence, there are only 9 input pads in this circuit; the address signals A-D, positive and negative supply voltages, ground, pixel voltage and an enable signal for the display presentation. On the contrary, no display is integrated with the decoder circuit in Fig. 4b, which adds the requirement of 14 additional output pads to enable the possibility to characterize the circuit during operation; 7 output pads for the respective segment counter electrodes and 7 pads for the respective segment pixel electrodes. There are 14 contact pads on the input side of the decoder circuit shown in Fig. 4b, however, only 9 of these pads are used for the actual input signals (the same signals as for the decoder circuit shown in Fig. 6), 4 pads are used as probe pads to measure the voltage output levels after inverting ($A_{\text{INVERSE}}-D_{\text{INVERSE}}$) the input signals A-D. The final pad is used as an extra GND connection to simplify routing.

The authors did not follow up my comment regarding the combination of shift register, BCD and display. I understand the role of each circuit, but it would have added value to see a monolithically integration of the 4-7 BCD (Fig. 4b) and the shift register (Fig. 5c) with the BCD.

I understand the justification not to use a Si chip in the demonstration.

Both the decoder and the shift register circuits are dedicated towards output devices, to e.g. enable an update sequence of a peripheral display device. The shift register provides a minimum of output terminals (one clock signal and one data input signal) required from the addressing electronics, e.g. a

Si-chip, and to combine this with a 4-7 decoder would make no sense. Maybe the reviewer is referring to another type of printed circuit capable of selecting one out of several electronic devices connected to the input terminals of the Si-chip, which then interprets the signals and finally addresses a printed decoder or shift register at the output terminals to update the display. Such a system solution, however, would not add value to the sophistication of the circuits already demonstrated in the current manuscript.

Methods / Manufacturing for contact materials comment:

The authors should include this explanation with references to the reduction front issue in the paper.

The reduction front issue is described in the main text of the manuscript, and a reference to this section has now been added in the Methods section.

Methods / Manufacturing for dielectric patterning comment:

I thank the authors for the clarification. The explanation is clear but it does not appear in the text, neither is this layer shown in the cross-section of the device.

From the text sentence “An insulating layer of 5018 ink is then deposited and UV-cured to define the area of the subsequently deposited gate electrolyte” the reader cannot know if the dielectric is patterned on the gate or around it (which is the case according the extra explanation provided by the authors).

The dielectric layer has now been added to the cross-section image of the OECT.

Other small comments:

Abstract:

- *It is not clear what the authors mean with the “former” and the “latter” devices.*
- *The sentence: “thus enabling few Si-output terminals to drive monolithically integrated all-printed 7-segment electrochromic displays” is misleading since it looks like the authors have demonstrated this interconnection*
- *“driving protocols” -> what are those in the paper?*

The abstract has been amended in revision taking into account the comments above.

Page 4

- *“This was obtained by increasing the margins of the voltage output levels” -> what from which value?*

This sentence has now been amended in revision.

- *“the minimized voltage strain implies...”-> minimizes from which value? Justify the selection of +/- 5 V*

We are grateful for this comment. In fact, the word ‘minimizing’ is confusing and we do not have evidence for that the voltage strain actually has been minimized; this has now been replaced with ‘lower’ instead.

Page 6

- *Explain the following in Fig. S6: “Additionally, the chosen values imply that the resistor ladder is tolerant to ~15% of resistance variation of the all-printed resistors (Fig. S6)”*

This is a good comment, and we agree that this statement was rather confusing in the previous version of the manuscript. This has now been replaced with ‘a resistance variation of 10 % would result in an output voltage variation of ~1 %’, to emphasize that the chosen resistor values allow variations in the screen printing process without affecting the output voltage levels. A revised Supplementary Figure 6 to support this statement in also provided in revision.